# Three-Dimensional Finite Element Analysis of Stress Distribution in a Tooth Restored with Full Coverage Machined Polymer Crown

**Azeem Ul Yaqin Syed** [1,2,*], **Dinesh Rokaya** [3,*], **Shirin Shahrbaf** [1] and **Nicolas Martin** [1]

1   Academic Unit of Restorative Dentistry, The School of Clinical Dentistry, University of Sheffield, Claremont Crescent, Sheffield S10 2TA, UK; s.shahrbaf@sheffield.ac.uk (S.S.); n.martin@sheffield.ac.uk (N.M.)
2   Department of Prosthodontics and Dental Implantology, College of Dentistry, King Faisal University, Al Ahsa 31982, Saudi Arabia
3   Department of Clinical Dentistry, Walailak University International College of Dentistry, Walailak University, Bangkok 10400, Thailand
*   Correspondence: azeem.ajaz@gmail.com (A.U.Y.S.); dinesh.ro@wu.ac.th (D.R.); Tel.: +44-775-110-7182 (A.U.Y.S.)

**Abstract:** The effect of a restored machined hybrid dental ceramic crown–tooth complex is not well understood. This study was conducted to determine the effect of the stress state of the machined hybrid dental ceramic crown using three-dimensional finite element analysis. Human premolars were prepared to receive full coverage crowns and restored with machined hybrid dental ceramic crowns using the resin cement. Then, the teeth were digitized using micro-computed tomography and the teeth were scanned with an optical intraoral scanner using an intraoral scanner. Three-dimensional digital models were generated using an interactive image processing software for the restored tooth complex. The generated models were imported into a finite element analysis software with all degrees of freedom concentrated on the outer surface of the root of the crown–tooth complex. To simulate average occlusal load subjected on a premolar a total load of 300 N was applied, 150 N at a buccal incline of the palatal cusp, and palatal incline of the buccal cusp. The von Mises stresses were calculated for the crown–tooth complex under simulated load application was determined. Three-dimensional finite element analysis showed that the stress distribution was more in the dentine and least in the cement. For the cement layer, the stresses were more concentrated on the buccal cusp tip. In dentine, stress was more on the cusp tips and coronal 1/3 of the root surface. The conventional crown preparation is a suitable option for machined polymer crowns with less stress distribution within the crown–tooth complex and can be a good aesthetic replacement in the posterior region. Enamic crowns are a good viable option in the posterior region.

**Keywords:** computer-aided design and computer-aided manufacturing; crown; dental ceramics; finite element analysis; micro-computed tomography; resin cement; von Mises stress

## 1. Introduction

The teeth and their supporting structures are under constant load in the form of chewing and biting forces. These forces in turn produce stresses within the complex tooth structure. Enamel is very susceptible to fractures as it is brittle in comparison to resilient dentine with low stiffness polymers [1]. The enamel dentine complex is of vital importance in maintaining and sharing load applications. It is well reported in the literature that any damage to tooth tissue requires restoration but the preservation of tooth tissue becomes critical when the gross amount of tooth structure has been lost [1]. The uses of dental restorations with more aesthetic and biologic properties have been long established and treatments with crowns with more life-like appearance [2]. The teeth have very different properties and behave differently under stress when naturally intact, compared to the crown–tooth complex [3]. In terms of resemblance to natural tooth structure, all-ceramic

restorations lacking metal substructure are most widely accepted due to their color and light translucency. More recently, metal-free polymer crowns have been introduced with better aesthetics. These polymer-based ceramic materials have emerged with properties that offer a new dimension to various applications in indirect restorations. There are many in vivo and in vitro studies that have evaluated the performance of dentine bonded crowns both in terms of fracture strength and clinical performance [4]. These polymer-based restorations are indeed clever manipulation of material with increased fracture strength when bonded to tooth tissue [5]. The longevity of restorations is purely dependent on several factors and the most important factor is the marginal fit of the restoration along with its internal adaptation to bond to the tooth structure [6]. Recently, computer-aided design and computer-aided manufacturing (CAD-CAM) technology can be used to fabricate ceramic restorations of less than 0.5 mm thickness [7]. Minimally invasive occlusal veneers made of ceramic and hybrid materials replaced the conventional crows for correction of occlusal tooth wear [7]. The most important mechanical property in material under load is its elastic modulus, which also defines its stress-strain relationship [8]. Many techniques have been implemented for stress analysis within dental core structures, such as 3D photoelasticity methods, 3D finite element, reflection microscopy, and stress-strain gauge techniques [9–11]. The materials being used in restorative dentistry for restorations have stiffness properties dissimilar to a natural tooth. Therefore, due to this mismatch of mechanical properties between the materials and tooth structure, problems arise when a restored tooth is subjected to loading conditions such as biting forces, chewing forces, and different parafunctional forces related to habits of the patients especially in the posterior region of the mouth [12]. When the restored crown–tooth complex is subjected to loads, the material with higher elastic modulus takes on the most wrath, as the dissipation of stresses generated does not spread evenly to the underlying tooth tissue. Therefore, cracks propagate within the ceramic structure, leading to failure in terms of chipping or complete fractures. Ceramics in turn undergo catastrophic failures due to this mismatch phenomenon [13]. For complex structures such as restored teeth, an accurate model is required to achieve appropriate data. As such, 2D models would not deliver good results [14].

Finite element analysis (FEA) is a more intriguing innovation technique to evaluate the stress-strain relationship and has been applied widely in dentistry [14–22]. FEA uses computerized data and is a numerical method of analyzing stress distribution. This is with the help of FEA that simultaneous deformations can be predicted and evaluated in various components of a restoration. In a complex 3D structure, FEA has made it possible to calculate the stress distribution within the structure by dividing the structure into tiny shapes called elements, and their ends meet to form nodes [22]. FEA, being a numerical approach to analyze stress distribution, is the most repeatedly used method for in vitro studies [23]. To obtain an accurate 3D model, a micro-computed tomography (micro-CT) scan can be used to generate the data [24]. This is helpful and is used because it saves time and reduces technical complexities.

Dawood et al. [18] evaluated the biomechanical and thermal behavior of a preparation design as a conservative treatment option that aims to preserve both gingival and tooth health structures using a finite element analysis with non-preparation and conventional designs. They found that the preparation design geometry affects the long-term success of laminate restoration and the proposed design yields more uniform and appropriate stress distributions than other techniques.

Similarly, another study studied the stress distributions in a finger bone using a dental implant and the loading force was applied along the long axis of the implant using finite element analysis [17]. They found that the maximum stress was located at the head of an abutment screw. The minimum stress was in the apical third of the implant fixture. The weakest point was calculated by the safety factor which was in the spongy bone at the apical third of the fixtures. Hence, to achieve long-term success, the designers of implant systems must confront biomaterial and biomechanical problems including in vivo forces on implants, load transmission to the interface, and interfacial tissue response.

The distribution of load and stresses generated within the crown–tooth complex is very different from to intact tooth compound. Therefore, this study aimed to investigate and evaluate the effect of stresses generated in a hybrid dental ceramic crown in the posterior region of the mouth, fabricated with CAD-CAM technology for a conventional tooth preparation design with the help of the FEA method.

## 2. Materials and Methods

### 2.1. Preparation of Bonded Crown–Tooth complex

Two extracted human maxillary second premolars for orthodontic purposes were carefully selected and mounted (Figure 1A). The inclusion criteria for teeth to be included are as follows: sound tooth structure both coronally and apically, without any defects, restoration free, permanent teeth, non-root-filled teeth, and the exclusion criteria; damaged natural tooth crown (caries, fracture, restored), severe attrition/wear, cracks, root-filled, immature apices, or root formation. The teeth were scanned with an optical intraoral scanner using an intraoral scanner using CEREC powder (VITA Zahnfabric, Bad Sackingen, Germany). Two teeth were selected so if in case any fracture or damage happened to a tooth while the study continued.

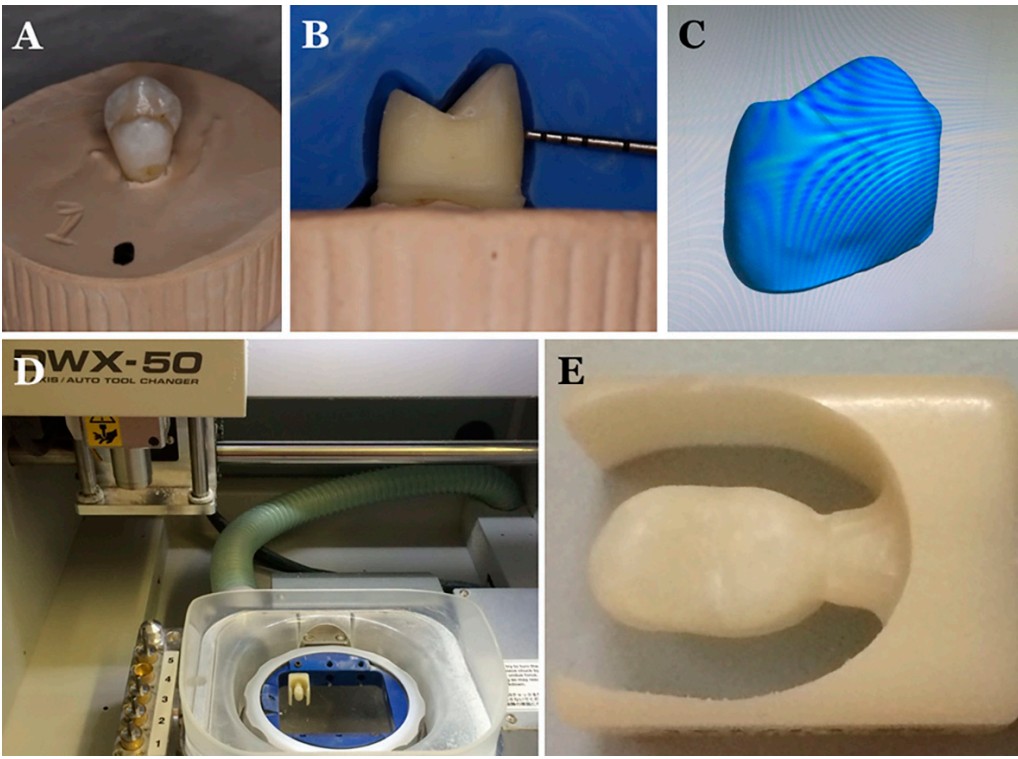

**Figure 1.** Tooth preparation and crown fabrication. (**A**) Mounted sound premolar. (**B**) Prepared tooth. (**C**) Crown designed from dental computer aided design and computer aided manufacturing software (exocad Gmbh, Darmstadt, Germany). (**D,E**) Crown milled from Vita Enamic® (VITA, Zahnfabrik, Germany) using Roland DWX-50 milling machine (Whip Mix GmbH, Louisville, KY, USA).

Both teeth were prepared according to the Vita®Enamic (VITA Zahnfabrik, Bad Sackingen, Germany) manufacturer's guidelines (Figure 1B), as shown in Table 1 [24,25]. The occlusal morphology of the scanned teeth was used for the machined polymer ceramic crowns. The restorations were designed using dental CAD software (exocad Gmbh, Darmstadt, Germany) (Figure 1C). The "anatomic crown mode" was selected to design the crown and "wax-up mode" was selected to copy the pre-scan anatomy/morphology so that the original morphology with all similar dimensions could be obtained. The thickness of the crown was standardized using the software and 50 μm was used as a spacer for the cement.

The STL files were saved and transferred using a flash drive to the third party milling machine. We used Roland DWX-50 (Whipmix, U.S.A) milling machine (Figure 1D) for preparation of Vita Enamic® (VITA Zahnfabrik, Bad Säckingen, Germany) crowns (Figure 1D). Cementation of Vita Enamic® (VITA Zahnfabrik, Bad Sackingen, Germany) crowns were made using resin cement (Rely X U200, 3M ESPE Dental Products, St. Paul, MN, USA) with the following criteria: good crown fit, good integrity of the margins, and good crown morphology having fissures and occlusal anatomy [24].

**Table 1.** Tooth reduction parameters for Vita®Enamic (VITA, Zahnfabrik, Germany) manufacturer's guidelines [24,25].

| Tooth Reduction | Vita Enamic® (VITA, Zahnfabrik, Germany) Manufacturer's Protocol |
|---|---|
| Occlusal reduction | Should be at least 1.5 mm |
| Bucco-labial or linguo-palatal reduction | At the bottom of the fissure: at least 1 mm, In the area of the cusps at least 1.5 mm |
| Margin all around | 0.8 to 1.5 mm |

The external surface of the crown was polished using a polishing set from a ceramics kit. The teeth were cleaned for any debris through a slurry of pumice and water. The internal crown surface was sandblasted using 50 μm $Al_2O_3$ using 1 bar blasting pressure. The adhesive (primer/bond) (Scotchbond™ Universal, 3M ESPE Dental Products, St. Paul, MN, USA) was applied to the prepared tooth for Vita® Enamic, (VITA, Zahnfabrik, Germany). The tooth was etched with 32% phosphoric acid gel for 30 s. Then, the bond, Scotchbond™ Universal, was applied and the restoration was seated. Then, the adhesive resin cement (Rely X U200, 3M ESPE Dental Products, St. Paul, MN, USA) was applied to the inner crown surface and the restoration was inserted and self-cure technique implied. The restorations fit/seating were standardized using the universal test machine (Lloyds Instrument Model LRX) with a load of 40 N for 3 min.

### 2.2. Finite Element Model Preparation

To generate a 3D model, micro-CT scans of the restored specimens were made. Scanning was performed using a high-resolution micro-CT scanner SkyScan 1172 (SkyScan, Aartselaar, Belgium). After scans, micro-CT reconstruction was conducted to shrink the number of output data. This volumetric reconstruction was carried out using Skyscan's NRecon® software. A cross-section of slices was generated through the scanned sample. NRecon® software was used to reduce errors by optimizing alignment. All unwanted data was removed, and beam hardening was done along with image sharpening, wherein any ring artifacts were corrected. This data was cropped using the cropping tool CTAn® analyzer (SkyScan, Aartselaar, Belgium). The only region of interest was included using the ROI tool. All images were reduced to a file size of fewer than 1 GB and reconstructed and converted to bitmap (.bmp) files.

To obtain a 3D model from 2D images via a micro-CT scan, a medical imaging software Mimics® 13.1 (Materialise NY, Leuven, Belgium) was used (Figure 2A). The object was seen in 3 views on the screen sagittal, coronal, and axial. Default x and y line axis represented the horizontal and vertical 2D of the body, respectively, and was adjusted within the frames by slight adjustments (Figure 2B). To create a mask of the object the first step was "thresholding". Thresholding visualizes the tooth in separate components; for example, a crown, cement, enamel, dentine, and pulp could be separately seen as masks. Segmentation was used to separate any component from each other. It allows for the creation of an accurate individual mask. The boolean operation was used to add or remove masks from each other; for example, crown and enamel masks were removed for a dentine mask. A boundary condition (zero displacements) was defined at all nodes of the teeth that were constrained in all directions (X, Y, and Z). Then, the coordinate axes were directed, as shown in Figure 2. In the end, the calculated mask option was used to generate a 3D model after the final optimization of each layer (Figure 2C).

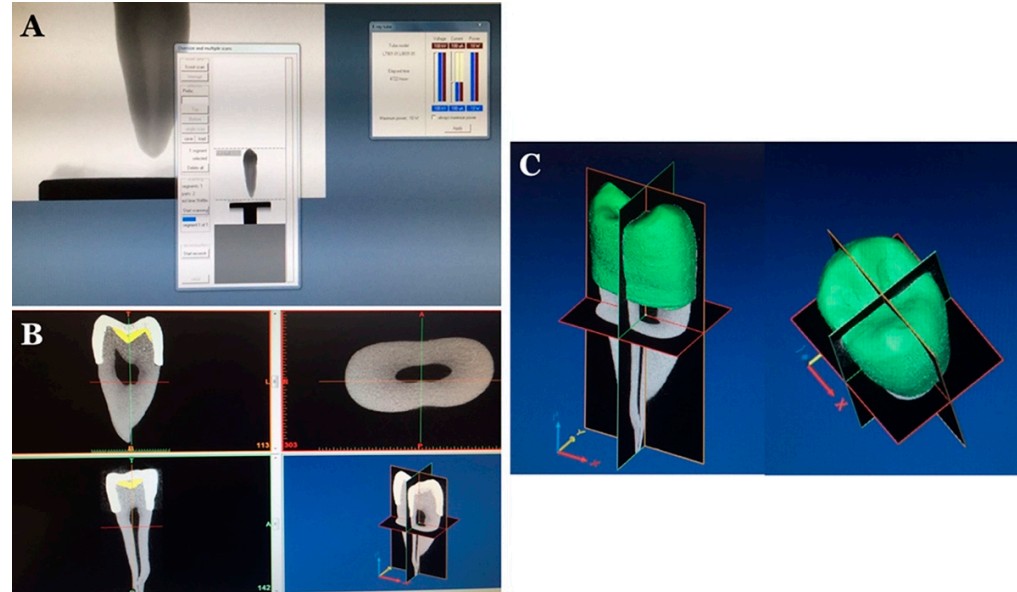

**Figure 2.** The 3D model generation. (**A**) Micro-Computed Topography images of the restored specimen. Scanning was performed using a high-resolution micro-CT scanner SkyScan 1172 (SkyScan, Aartselaar, Belgium). (**B**) The 2D model of the tooth with crown. (**C**) The 3D model generated from 2D images secured from micro-CT scan from Mimics® 13.1 (Materialise NY, Leuven, Belgium).

The final 3D models for each component were exported in a standard triangle language (STL) format which consists of a mesh of small objects in a shell format (Figure 3). This high-quality mesh was created using models Hypermesh® (HyperWorks software, Version 11.0, Altair Engineering, Inc., Troy, MI, USA).

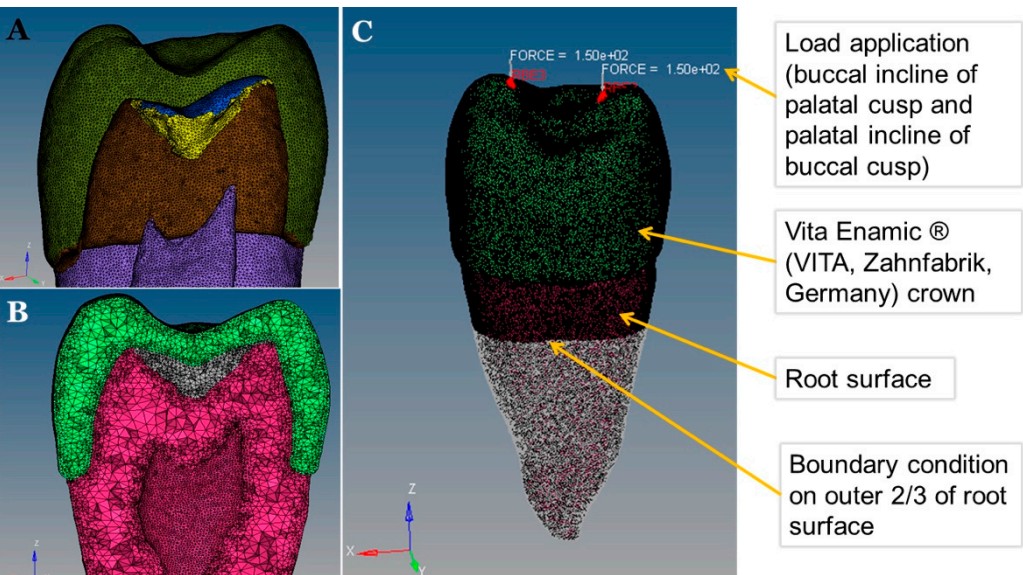

**Figure 3.** Model mesh. (**A**), the 2D model mesh. (**B**), the 3D mesh generation of conventional crown–tooth complex from Hypermesh® (HyperWorks software, Version 11.0, Altair Engineering, Inc., Troy, MI, USA). (**C**), the 3D model showing the crown and root, as well as the load application.

### 2.3. Material Associated Properties

In an elastic setting, the stress-strain state of the test tooth is determined by the modulus of elasticity and Poisson's ratio. Table 2 provides the values of elastic modulus and Poisson's ratio of the materials used in this study.

**Table 2.** Elastic modulus and Poisson's ratio of the materials used in this study.

| Materials | Modulus of Elasticity (MPa) | Poisson's Ratio | Reference |
|---|---|---|---|
| Vita Enamic® (VITA Zahnfabrik, Bad Sackingen, Germany) | 30,000 | 0.23 | [25] |
| (Rely X U200, 3M ESPE Dental Products, St. Paul, MN, USA) | 7700 | 0.30 | [26] |
| Enamel | 84,100 | 0.30 | [27] |
| Dentine | 18,600 | 0.31 | [28,29] |

### 2.4. Interface Association

The current study was considered to have perfect bonding and rigid conditions without any slip between components (crown, cement, dentine, and enamel). All the structures and materials used in the models were considered linear, elastic, homogeneous, and isotropic to reduce the complexity of the structures [30,31].

### 2.5. Boundary Condition and Load Application on the FE Model

Once 3D meshes were obtained, a boundary condition (zero displacements) was set to the external surface of the root [31,32]. The stress state in any object can only be investigated by the application of the load. In this study, we applied load at points at the outer surface of the crowns. The points are a group of nodes on the occlusal surface at the buccal inclination of the palatal cusp and the palatal inclination of the buccal cusp. To simulate the average occlusal load subjected on a premolar, a total load of 300 N was applied, 150 N at each loading point [30].

In the present study, the boundary condition for the tooth constrained each of the nodes located at its external part for all three degrees of freedom. This is a realistic boundary condition capable of providing an optimum prediction of stress state in the cemented machined all-ceramic crown [30]. A series of highly refined and accurate volume meshes were constructed to replicate the natural crown–tooth complex.

The finite element method was characterized by the computational algorithms using the variational formulation, a discretization strategy, one or more solution algorithms, and post-processing procedures. Examples of the variational formulation are the Galerkin method, the discontinuous Galerkin method, Wang–Landau, GMRES, mixed methods, etc. In this study, we assumed that the computer software used a mixed algorithm [33].

### 3. Results

von Mises stress was calculated for the crown–tooth complex. The peak value for von Mises stress generated within the different layers can be seen in Table 3. The von Mises stress dissipated in the different layers, showing a linear fashion of distribution amongst the crown, dentine, and cement layers. Most stresses were seen in the crown followed by dentine and the least in the cement layer (Figure 4). At the load application points, the polymer hybrid ceramic crown showed greater values for von Mises stress (Figure 5). For the cement layer, the stresses were more concentrated on the buccal cusp tip. In dentine, stress was more on the cusp tips and coronal 1/3 of the root surface. Only one tooth in the FEA analysis was carried out to save time, as it is a very precise tool that gives precise stress distribution around the tooth crown structures.

**Table 3.** The von Mises stress values (MPa) in the crown–tooth complex for Vita Enamic® (VITA, Zahnnfabrik, Germany) crown.

| Vita Enamic Crown layer Von Mises Stress (MPa) | Dentine Layer Von Mises Stress (MPa) | Cement Layer Von Mises Stress (MPa) |
| :---: | :---: | :---: |
| 9.43 | 7.80 | 3.34 |

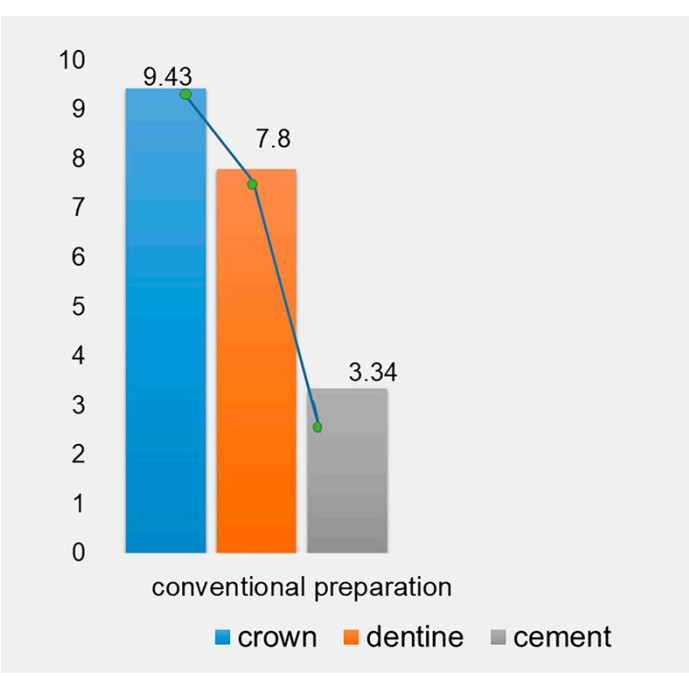

**Figure 4.** Peak von Mises stress values in MPa of Vita Enamic® crown, dentin, and cement in conventional preparation design.

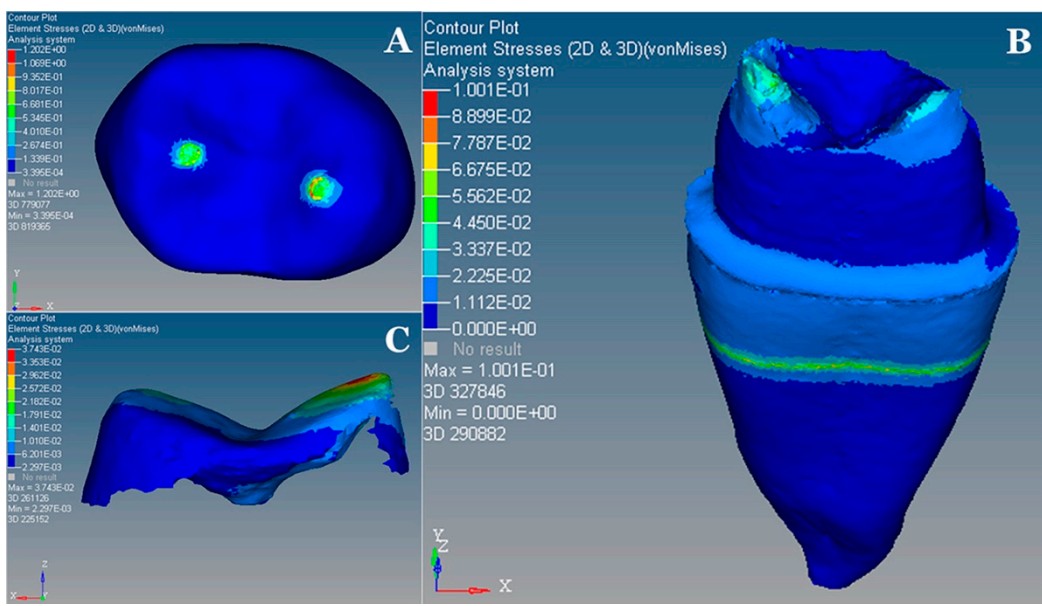

**Figure 5.** Stress generation on the crown (**A**), stress generation in cement layer of conventional preparation design (**B**), and stress generation in the dentin of conventional preparation design (**C**).

## 4. Discussion

Tooth preparations and removal of sound tooth structure generate varying results and no single mechanical test can give statistical results that can be relative to tooth loading

encountered in the mouth. A particular system evolves a particular tooth preparation with manufacturer guidelines. Metal ceramics, all ceramics, and a more recent innovation are of polymer-based ceramic crowns. These crowns have preparation guidelines that require a hefty amount of tooth preparation. Thus, we rendered an overall weak tooth structure under a biological cost that is unmatched by the natural tooth. Therefore, we need to address the survivability of the crown–tooth complex rather than an individual entity.

In this study, to evaluate the stress pattern generated in a tooth restored with a machined polymer-based hybrid ceramic crown, a 3D model was created. A conventional preparation design model was chosen according to the manufacturer's guidelines. The tooth and Vita®Enamic (VITA Zahnfabrik, Bad Sackingen, Germany) manufacturer's guidelines crown complex underwent loading conditions of 150 N at a buccal incline of the palatal cusp and palatal incline of the buccal cusp (total 300 N). The stress patterns were simulated using 3D finite element study [34].

In the FE analysis, some assumptions were made, which are relayed as follows: (i) there was a perfect bond between the machined hybrid dental ceramic crown, the dentine, and the cement layer; (ii) all layers of the structures were assumed to be linear, elastic, and isotropic; and (iii) enamel was considered as homogenous and isotropic [31]. For load distribution, it has been reported that oblique loads are of much more clinical relevance and generate more stress than vertical loads [35]. In this study, to compare the stresses generated within a crown–tooth complex, axial loading was opted for, as it is more effective and the point of application of load can be promptly reproduced [36–38]. FE analysis can be only done with the establishment of boundary conditions and, in this study, we created it on the outer 2/3 surface of the tooth root which is where most of the movement occurs during loading conditions or occlusal forces in the mouth [31].

According to the data obtained for von Mises stress, the manufacturer's preparation design showed the variation of stress pattern amongst all three layers of crown, dentine, and cement. There were high stresses in the crown, dentine, and less in cement, which is in agreement with Shirin et al. (2013), as they compared different lute parameters, which we did in this study. Oyar et al. [39] investigated different occlusal designs and documented more stress in the anatomical design than in the flat. Moreover, they reported less stress in the dentine layer for the anatomic design. This stress they reported may be due to the amount of reduction at 2 mm. In this study, we had an occlusal preparation of 1.5 mm, and results for this design were in accordance with the Shirin et al. study results for von Mises stress [39]. It is of importance to note that Shahrbaf et al. [30] demonstrated similar results for their anatomical design for cement and dentine von Mises analysis when a cement of elastic modulus of 8 GPa was used. In this study, the cement used had an elastic modulus of 7.7 GPa.

In this study, we wanted to evaluate the manufacturer's design of preparation for Vita® Enamic crowns and their stress distribution in the posterior region while loading. For a better insight into stresses generated in the crown–tooth complex, it is of utmost importance to consider several other factors such as milling units used, the minimal amount of crown produced without any flaws, the gap interface between the crown margin and the tooth, and the elastic modulus of different types of cement used. In this study, we faced challenges with modeling due to the segmentation of different layers within a very close thresholding range. The cement layer was uneven and did not follow the uniform pattern throughout. That said, a 'growing a threshold region' and 'boolean operation' were not used but could have helped in better-computed models. Moreover, this could be explained in terms of more cement was present at the occlusal interface just due to the fact of creating a 50 μm space within the CAD-CAM design. It is also of importance to note that the CAD-CAM does not have the ability to mill the internal anatomy to the desired preparation and due to this there is sufficiently more gap present on the occlusal surface of anatomical design preparations compared to the marginal gap that can be created according to the desired outcomes. Therefore, the CAD-CAM milling burs lack the proper internal adaptation of the occlusal surface, leaving far more space for the cement [7]. The

axial walls hardly had any cement attached since milling was done under a close marginal fit of the crowns. This aspect was important to achieve a good fit of the crowns and to avoid thin or weak crowns at the margins. For the prepared tooth we had enough thickness at the margins, but a close fit was chosen to avoid the possibility of bulky crowns. Therefore, due to CAD-CAM milling deficiencies, we could not achieve uniform cementation as would have been expected. Moreover, there might be some preparation discrepancies along with cuspal tip interferences for cementation as it followed the occlusal anatomy. Moreover, a possibility of cement heterogeneity with a thick consistency and in fewer amounts would have led to an uneven spread throughout the margins but more along the occlusal surface. Shahrbaf et al. [30] also showed a flat design with a more homogenous cement thickness all around rather than an anatomical design where more cement was found at the occlusal surface.

More stress was seen in the crown and has also been reported by Shahrbaf et al. [30] and Oyar et al. [39] for the anatomic design. Dentine showed stress generation along the coronal 1/3 of the root surface, which is where most stresses are dissipated even in loading/masticatory/chewing conditions in the mouth (also reported by Ausiello et al.) [40]. Within the cement layer, more stress was generated in the occlusal surface rather than a uniform spread around the areas of the defect since the thick cement layer had occlusal distributed the load. Lower elastic modulus also plays an important role in the dissipation of stresses, but this was not the case in this study probably due to the uneven distribution of the cement.

Some limitations exist in this FE study, such as nonlinear simulation of periodontal ligament properties, plastic and viscoelastic behaviors in materials, and tooth-to-tooth contact analyses. However, there are difficulties such as the dynamic behavior of the periodontal ligament, which is an aspect to be considered. Yet due to its complex structure, the exact mechanical properties of PDL are considered to be poorly understood. Importantly, the different aspects like viscoelastic behavior and cyclic loading have not been performed, which can be done to compare results in future studies.

## 5. Conclusions

With all of the above information, this study has been performed in a restored crown–tooth complex with the use of a 3D model and FE analysis on a tooth preparation according to manufacturer's guidelines. Three-dimensional finite element analysis showed that the stress distribution was more in the dentine and least in the cement. The conventional design showed good stress dissipation as contemplated and further Vita® Enamic crowns are a viable option in the posterior region. Still, caution is needed to stipulate the computational study in mouth and more studies are required to validate these results in the future.

**Author Contributions:** Conceptualization, A.U.Y.S., S.S., and N.M.; methodology, A.U.Y.S., S.S., and N.M. software, A.U.Y.S., S.S., and N.M.; validation, A.U.Y.S., S.S., D.R., and N.M.; formal analysis, A.U.Y.S., S.S., and N.M.; investigation, A.U.Y.S.; resources, A.U.Y.S. and D.R.; data curation, A.U.Y.S., D.R., S.S., and N.M.; writing—original draft preparation, A.U.Y.S. and D.R. writing—review and editing, A.U.Y.S. and D.R.; visualization, A.U.Y.S. and D.R.; supervision, N.M.; project administration, S.S. and N.M.; funding acquisition, A.U.Y.S. All authors have read and agreed to the published version of the manuscript.

**Funding:** This research received no external funding.

**Institutional Review Board Statement:** Not applicable.

**Informed Consent Statement:** Not applicable.

**Data Availability Statement:** The data presented in this study are available on request from the corresponding authors.

**Acknowledgments:** I would want to thank the ALMIGHTY for giving me strength, dedication, and patience throughout this project until its completion. I would want to show my sincere gratitude to my supervisors involved in this project. I am indebted for the continuous support and guidance

**Conflicts of Interest:** The authors declare no conflict of interest.

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
