# Peer review of "Three-Dimensional Finite Element Analysis of Stress Distribution in a Tooth Restored with Full Coverage Machined Polymer Crown"

_applsci, doi:10.3390/app11031220_

Round 1

Reviewer 1 Report

REVIEW

on article

Three-dimensional finite element analysis of stress distribution in a tooth restored with full coverage machined polymer crown

Azeem Ul Yaqin Syed, Dinesh Rokaya, Shirin Shahrbaf and Nicolas Martin

SUMMARY.

The article is devoted to the study of the stress-strain state of a mechanically processed hybrid dental ceramic crown-tooth complex. A numerical study was performed using a 3D finite element analysis of the stress state effect of a treated hybrid dental ceramic crown.

The molar tooth of a human is a complex heterogeneous structure. The enamel covering the tooth is very sensitive to stress and is vital for maintaining human health. In this regard, the aim of the study related to the study and assessment of the effect of stresses arising in a hybrid dental ceramic crown made using the innovative CAD-CAM technology is an urgent scientific problem and will attract the attention of readers.

The authors extracted two molars. They were scanned using an optical intraoral scanner and a 3D digital model was built, which was investigated by numerical methods.

The article is well-structured and logical. The reference list contains 33 items.

Also, the strong point of the article is that the authors in the Discussion compare their results with the data obtained by other authors. This will surely attract the attention of readers.

COMMENTS.

  1. Line 22-23. The sentence needs to be rephrased. In a geometrically complex object with different properties of layers, the stress-strain state cannot be linear, but rather piecewise-linear.
  2. Line 135. There are extra points after the word “max”.
  3. Line 168-169. I would recommend the authors to formulate the problem of solid mechanics. The tooth, as an object of mechanical research, occupies the area O in space. We direct the coordinate axes, as shown in Fig. 2. The origin is set in ... And so on.
  4. Line 155. "Only" - The only
  5. Line 156. "...less" - fewer
  6. Line 184-185. The proposal needs to be rephrased; it sounds trite. For example: "In an elastic setting, the stress-strain state of the test tooth is determined by the modulus of elasticity and Poisson's ratio."
  7. Line 186. If these are references in Table 2, then the references must be in square brackets.
  8. Section 2.5 should be expanded. How many elements and how many degrees of freedom did the model have? What elements simulated the mesh? What algorithms were used in the calculation?
  9. Applying a load at a point theoretically creates infinite stresses. How correct is this statement and why not simulate contact over the surface?
  10. Von Mises stresses make sense when considering the possibility of plastic deformation. Have you considered such a task? What is the limit value for tooth load? What are the features of the deformed state of the tooth?

In general, the article solves interesting and relevant problems and is of scientific interest. However, there are many ambiguities in the article, so I recommend the article for publication after major corrections.

Author Response

Thank you for your positive comments. All your comments have been addressed. Corrections in the Manuscript are highlighted in Yellow color.

Point 1: Line 22-23. The sentence needs to be rephrased. In a geometrically complex object with different properties of layers, the stress-strain state cannot be linear, but rather piecewise-linear.

Response 1: The sentence is rephrased.

Point 2: Line 135. There are extra points after the word “max”.

Response 2: Corrected.

Point 3: Line 168-169. I would recommend the authors to formulate the problem of solid mechanics. The tooth, as an object of mechanical research, occupies the area O in space. We direct the coordinate axes, as shown in Fig. 2. The origin is set in ... And so on.

Response 3: This part is explained more according to the recommendation by the reviewer.

Point 4: Line 155. "Only" - The only

Response 4: Corrected.

Point 5: Line 156. "...less" - fewer

Response 5: Corrected.

Point 6: Line 184-185. The proposal needs to be rephrased; it sounds trite. For example: "In an elastic setting, the stress-strain state of the test tooth is determined by the modulus of elasticity and Poisson's ratio."

Response 6: Corrected. Thank you for your better recommendation.

Point 7: Line 186. If these are references in Table 2, then the references must be in square brackets.

Response 7: References are added in the square bracket.

Point 8: Section 2.5 should be expanded. How many elements and how many degrees of freedom did the model have? What elements simulated the mesh? What algorithms were used in the calculation?

Response 8: In the present study, the boundary condition for the restored tooth has been constraining each of the nodes located at the most external part of the cortical tooth for all three degrees of freedom. A boundary condition (zero displacements) was defined at all nodes of the teeth that were constrained in all directions (X, Y, and Z). This is a realistic boundary condition capable of providing an optimum pre-diction of stress state in the cemented machined all-ceramic crown. A series of highly refined and accurate volume mesh was constructed replicating the natural crown–tooth complex. (Page 5-6)

The finite element method was characterized by the computational algorithms using variational formulation, a discretization strategy, one or more solution algorithms, and post-processing procedures. Examples of the variational formulation are the Galerkin method, the discontinuous Galerkin method, Wang–Landau, GMRES, mixed methods, etc. In this study, we assumed that the software used mixed algorithm.

Point 9: Applying a load at a point theoretically creates infinite stresses. How correct is this statement and why not simulate contact over the surface?

Response 9: We agree with the reviewer that applying a load at a point theoretically creates infinite stresses. In this study, we first simulate contact over the surface. Then, we applied load at points at the outer surface of the crowns. The points are a group of nodes on the occlusal surface at the buccal inclination of the palatal cusp and palatal inclination of the buccal cusp. To simulate average occlusal load subjected on a premolar a total load of 300 N was applied, 150 N at each loading point [24].

Point 10: Von Mises stresses make sense when considering the possibility of plastic deformation. Have you considered such a task? What is the limit value for tooth load? What are the features of the deformed state of the tooth?

Response 10: In this study, we applied load on the occlusal surface at the buccal inclination of the palatal cusp and palatal inclination of the buccal cusp. A total load of 300 N was applied on premolar, 150 N at each loading point [24].

Another thing, there is possibility of plastic deformation of tooth. We did not check the limit value for tooth load. This study was done to determine the effect of the stress state of the machined hybrid dental ceramic crown using three-dimensional finite element analysis. We have added these factors into our limitations such as nonlinear simulation of periodontal ligament properties, plastic and viscoelastic behaviors in materials, and tooth-to-tooth contact analyses. Future studies can be done considering these factors.

Point 11: In general, the article solves interesting and relevant problems and is of scientific interest. However, there are many ambiguities in the article, so I recommend the article for publication after major corrections.

Response 11: All corrections are done.

Reviewer 2 Report

Peer Review Report

Ms. Ref. No.: applsci-1087317

Title: Three-dimensional finite element analysis of stress distribution in a tooth restored with full coverage machined polymer crown

Authors: Azeem Ul Yaqin Syed, Dinesh Rokaya, Shirin Shahrbaf, Nicolas Martin

The subject presented in the manuscript is very interesting. However, the manuscript needs to be improved in my opinion. I recommend the paper for minor revision. The subject of the article is within scope of the journal. I believe that the authors will find below some suggestions, which will help them to improve their manuscript:

Major comments:

  • Do not introduce abbreviations in the abstract and keywords.
  • Introduction need to be extended considering and referring to the papers on the FEM applied for analysis of dental crown. In this way, the authors should highlight the novelty of the presented study in the area of the FEA for determination of stress distribution in dental crowns. Currently, the authors mentioned only some papers in the other parts of the manuscript. It should be definitively extended and included in Introduction.
  • Please provide more details on the mesh used – at least what was the number of elements used in this study.

Minor comments:

  • Please use spaces between numbers and units, e.g., in line 238 “300 N” instead of “300N” and “150 N” instead of “150N” or in Table 1. Please check it in the whole manuscript.

Conclusion:

The subject of the paper and the manuscript are very interesting. I recommend the manuscript for minor revision.

Author Response

Thank you for your positive comments. All your comments have been addressed. Corrections in the Manuscript are highlighted in Green color.

Point 1: Do not introduce abbreviations in the abstract and keywords.

Response 1: Abbreviations are changed to full form in the abstract and keywords.

Point 2: Introduction need to be extended considering and referring to the papers on the FEM applied for analysis of dental crown. In this way, the authors should highlight the novelty of the presented study in the area of the FEA for determination of stress distribution in dental crowns. Currently, the authors mentioned only some papers in the other parts of the manuscript. It should be definitively extended and included in Introduction.

Response 2: More related literatures are added in the Introduction. Novelty of the present paper is added.

Point 3: Please provide more details on the mesh used – at least what was the number of elements used in this study.

Response 3: Results are edited

Point 4: Please use spaces between numbers and units, e.g., in line 238 “300 N” instead of “300N” and “150 N” instead of “150N” or in Table 1. Please check it in the whole manuscript.

Response 4: The spacing is corrected through the manuscript.

Reviewer 3 Report

the current study employs FE modelling to simulate dental ceramic crown teeth stress states when being machined. The authors uses 3D imaging software and computerised tomography to create a restoration of the scanned teeth. The authors report on developed von mises stresses and stress distributions.

The abstract needs restructuring, it is not important to mention the software names but rather mention what has been done and what were the results.

It is not clear from the abstract what are the main findings in the study

Line 17 what does the authors mean by sophisticated? It is better to remove this word

Line 32 “a lot” the authors should check the manuscript for English and proof reading and use of scientific wording

The authors must avoid using we, our ..etc such as in line 36, 197 etc…

Lines 129-145 the authors should replace this bullet point style with a paragraph as this way it sounds like a manual or a set of instructions

Line 147 “were done..” check this sentence perhaps you mean were fabricated?

Figure 4 y axis missing units and description

Author Response

Thank you for your positive comments. All your comments have been addressed. Corrections in the Manuscript are highlighted in Pink color.

Point 1: The abstract needs restructuring, it is not important to mention the software names but rather mention what has been done and what were the results.

Response 1: Abstract is edited. Software names are removed. More details on method and results are added.

Point 2: It is not clear from the abstract what are the main findings in the study

Response 2: More findings are added.

Point 3: Line 17 what does the authors mean by sophisticated? It is better to remove this word

Response 3: ‘Sophisticated’ word is removed.

Point 4: Line 32 “a lot” the authors should check the manuscript for English and proof reading and use of scientific wording

Response 4: The word ‘a lot’ is removed. English correction is done throughout the manuscript.

Point 5: The authors must avoid using we, our ..etc such as in line 36, 197 etc…

Response 5: ‘We’ and ‘our’ words are removed from the manuscript.

Point 6: Lines 129-145 the authors should replace this bullet point style with a paragraph as this way it sounds like a manual or a set of instructions

Response 6: The bullet points are removed and made into paragraph.

Point 7: Line 147 “were done..” check this sentence perhaps you mean were fabricated?

Response 7: Corrected.

Point 8: Figure 4 y axis missing units and description

Response 8: Units explained.

Round 2

Reviewer 1 Report

All my comments were taken into account, studied and corrected. The article makes a good impression, has relevant content, and is of scientific interest. I recommend the article for publication.